# Excitation of Terahertz Spoof Surface Plasmons on a Roofed Metallic Grating by an Electron Beam

**DOI:** 10.3390/mi15030293

**Published:** 2024-02-21

**Authors:** Yongqiang Liu, Xutao Zhang, Yan Wang, He Cai, Jinhai Sun, Yong Zhu, Liangsheng Li

**Affiliations:** National Key Laboratory of Scattering and Radiation, Beijing 100854, China; 13240194055@163.com (X.Z.); wangyan2307@126.com (Y.W.); 18510165603@126.com (H.C.); jinhaisun@126.com (J.S.); 13810065549@163.com (Y.Z.); liliangshengbititp@163.com (L.L.)

**Keywords:** spoof surface plasmons (SSPs), roofed metallic grating, leaky wave SSP mode, terahertz radiation source, electron beam, high-power excitation

## Abstract

In this paper, both fundamental SSP modes on a roofed metallic grating and its effective excitation of the bounded SSP mode by an injected electron beam on the structure are numerically examined and investigated in the THz regime. Apart from the bounded SSP mode on the metallic grating with open space, the introduced roofed metallic grating can generate a closed waveguide mode that occupies the dispersion region outside the light line. The closed waveguide mode shifts gradually to a higher frequency band with a decreased gap size, while the bounded SSP mode line becomes lower. The effective excitation of the bounded SSP mode on this roofed metallic grating is also implemented and studied by using a particle-in-cell simulation studio. The output SSP power spectrums with various gap sizes by the same electron beam on this roofed metallic grating are obtained and analyzed. The simulation results reveal that the generated SSP spectra show a slight red shift with a decreased gap size. This work on the excitation of the SSP mode using an electron beam can benefit the development of high-power compact THz radiation sources by utilizing the strong near-field confinement of SSPs on metallic gratings.

## 1. Introduction

Spoof surface plasmon polaritons (SSPs) are a special kind of surface waves that propagate or localize on periodical textured metallic surfaces or related electromagnetic interfaces [1,2,3]. Due to their many excellent electromagnetic properties such as sub-wavelength wave guiding, enhanced light–matter interaction and versatile dispersion manipulation, etc., “spoof” plasmonics have attracted tremendous attention and become a frontier in both microwave and terahertz (THz) regimes [4,5]. SSP modes on various patterned metallic shapes have been investigated, including metal gratings [6], wedge structurse [7], cylindrical corrugated disks [8], “T”-shaped grooves [9], trapezoidal grooves [10], etc. Their fundamental guiding schemes such as strong near-field locations and low-loss propagations have been extensively pursued and demonstrated. Moreover, these novel waveguides and meta-structures have also been proposed for some functional devices and systems such as plasmonic lenses [11,12,13,14,15], on-chip filters and splitters [16,17,18], polarization converters [19,20], SSP antennas [21], OAM generators [22], etc. 

Among the various SSP-based devices and applications, their excitation on corrugated metasurfaces using various approaches is an important aspect and has also been largely studied in recent years. In order to realize an efficient dispersion momentum match to excite the SSP mode, various phase gradient reflective or transmissive metasurfaces have been proposed and demonstrated in the microwave band [23,24,25,26]. In these works, dispersion engineering of metasurfaces is key to excite the SSP mode with high-efficiency and versatile manipulations. In addition, the excited SSP mode can also be amplified by inserting some active circuit chips or other components [27], which can also be termed as “active plasmonic metamaterials”. Apart from the excitation or amplification mechanisms of SSP modes mentioned, high-power SSP emitters on meta-structure arrays using active photon or electron sources have also been largely demonstrated [28,29,30,31,32,33]. The working principles of the active excitation method rely on an effective dispersion match between the injected energy sources (photons or electrons) and SSP mode; thus, the interaction or amplification of the SSP mode can provide a new venue to obtain high-power radiation sources [29].

As an important SSP waveguide and interaction circuit, a metallic grating formed by single rectangular groove arrays has been proposed to develop free-electron-driven high-power radiation or amplifier sources [34,35,36,37]. It is shown that the SSP mode can be effectively excited and gradually amplified by an injected electron beam both on uniform and gradient meta-structures. These studies focus mainly on open space metallic grating structures (the inset in Figure 1a) and the interaction between an electron beam and the SSP mode is also simple. Some recent studies indicate that SSP properties and dispersion diagrams can be largely modified or tuned by introducing a closed metal plate in the vicinity of the metallic grating, namely on the roofed metallic grating (the inset in Figure 1b) [38,39,40,41,42,43,44,45,46,47]. Thus, the variation in SSP modes on this kind of roofed metallic grating with different gap sizes should be considered carefully. Additionally, the introduced metal plate may also influence the various plasmonic devices, including high-power radiation sources, based on the interaction between the SSP mode and injected electron beam [34,35,36,37,48,49], which needs to be studied in detail. 

According to the above-mentioned insufficient studies in the existing literature, we here propose the excitation of the THz SSP mode on a roofed metallic grating using an injected electron beam. The basic SSP dispersion diagrams on the open or roofed metallic grating are investigated and compared first. In addition, the influence of the gap size between the bare metallic grating and metal plate on the SSP mode is specifically studied. Also, the excitation of the bounded SSP mode by an injected electron beam on the roofed metallic grating is also implemented and modeled by using particle-in-cell simulations. The output power spectrums of the SSP mode with different gap sizes are also given and analyzed. The presented studies can provide a new path to develop compact THz radiation sources induced by an injected electron beam based on the roofed SSP waveguide.

## 2. Theoretical Model and Dispersion Theory

The proposed SSP wave excitation by an injected electron beam is based on a rectangular groove periodical metallic grating with an open space or closed boundary at the top, as schematically plotted in the inset of Figure 1a,b, respectively. The metallic grating has the same groove depth of *h*, groove width of a and period of *d*, respectively, as shown in the inset of Figure 1. A metal plate is introduced to this plasmonic waveguide and the distance of the metal plate from the grating surface is marked by *g* in the inset of Figure 1b [38,39,40,41,42,43,44]. We first consider SSP mode dispersion on the *x*-*z* plane, which is assumed to be the transverse magnetic mode (TM) along *z* and metal is set as the ideal conductor for the study [39]. The whole structure is divided into two different regions: under and above the grating surface, respectively. The initial point of the *x* axis is on the grating surface. The dispersion characteristics of the SSP mode for the open space or roofed metallic grating structure are studied using an effective medium method [40] or rigorous field expansion method [39,42], as mentioned previously. We here employ a simplified mode matching method to obtain its dispersion property on the metallic grating. For the open space metallic grating or closed metallic grating in Figure 1, the fields under the metallic surface are the same and can be expressed as homogeneous, based on the fact that the lattice period is usually shorter than the wavelength in the THz band [34,35,36,37,38,39,40,41,42,43,44,45]:
(1)$$E_{z}^{u}=A\sin k(x+h)e^{-j\beta_{0}md}$$

Also, its magnetic field can be obtained based on its relation to the TM mode:(2)$$H_{y}^{u}=A\frac{k}{j\omega \mu }\cos k(x+h)e^{-j\beta_{0}md}$$
where *A* is the unknown index and *md* (*m* = 0, 1, 2, 3…) is the axial distance, *k* = *ω*/*c* is the wave vector in free space, *μ* is vacuum permeability, *β*_0_ is the propagation constant of the fundamental mode, *ω* is the angular frequency and *j* is the imaginary unit. For the fields above the metallic grating, its formulism is different because of the different open space or roofed metallic plate boundary condition. In the bare metallic grating of Figure 1a, its fields can be expressed as:(3)$$E_{z}^{a}=\sum_{n=-\infty }^{\infty }B_{n}e^{k_{n}x}e^{-j\beta_{n}z}$$
(4)$$H_{y}^{a}=\sum_{n=-\infty }^{\infty }\frac{-j\omega \varepsilon_{0}}{k_{n}}B_{n}e^{k_{n}x}e^{-j\beta_{n}z}$$
where *B_n_* is the unknown index and the periodical harmonic mode along the propagation direction is considered as *β_n_* = *β*_0_ + 2*nπ*/*d*, while *k*^2^*_n_* = *β*^2^*_n_* − *k*^2^ (*n* = 0, ±1, ±2, ±3…) is the propagation constant along the *x* axis. The axial field *E_z_* above the grating surface decays exponentially along the *x* axis because of its infinite open space boundary condition.

If we consider the roofed metallic grating in Figure 1b, its fields above the metallic grating change into the following forms:
(5)$$E_{z}^{a}=\sum_{n=-\infty }^{\infty }B_{n}\sinh k_{n}(g-x)e^{-j\beta_{n}z}$$
(6)$$H_{y}^{a}=\sum_{n=-\infty }^{\infty }\frac{-j\omega \varepsilon_{0}}{k_{n}}B_{n}\cosh k_{n}(g-x)e^{-j\beta_{n}z}$$

It can be noted that the fields above the gating are closely related to the distance of the metal plate of *g*. Based on the rigorous tangential electric field continuity of *E_z_* and magnetic flow conservation of *H_y_* for one period in region I and II [36], the SSP dispersion expression can be obtained after tedious calculations, which eliminate the above unknown index (*A* and *B_n_*). Based on the field expressions of Equations (1)–(4), the SSP dispersion relation to the open space grating in Figure 1a is as follows:
(7)$$\sum_{n=-\infty }^{\infty }\frac{\sin c^{2}(\frac{\beta_{n}a}{2})}{k_{n}}=\frac{d}{a}\frac{1}{k}\cot(kh)$$

Also, the SSP dispersion expression on the roofed metallic grating shown in Figure 1b can also be calculated based on Equations (1), (2), (5) and (6) as follows:
(8)$$\sum_{n=-\infty }^{\infty }\frac{\sin c^{2}(\frac{\beta_{n}a}{2})}{k_{n}\tanh(k_{n}g)}=\frac{d}{a}\frac{1}{k}\cot(kh)$$

The function sin*c*(*β_n_a*/2) is sin(*β_n_a*/2)/(*β_n_a*/2) in the dispersion expression and the other symbols are the same as those in the above-mentioned field expressions. Our derivation processes are simplified compared to previous studies for similar roofed metallic gratings and include the effective medium method or rigorous field expansion method, which consider the SSP harmonic mode [38,39,40,41,42]. Furthermore, it can be noted that the SSP mode in Expression (8) on the roofed metallic grating can be simplified to Equation (7) if the gap size of *g* is infinite [39]. 

According to the above analytic SSP dispersion expressions in Equations (7) and (8), the dispersion lines of SSP modes within the first Brillouin zone are plotted and presented in Figure 1a,b, respectively. For the bare metallic grating with open space, there is only one well-known bounded surface mode of the green line under the light line. For the roofed metallic grating in Figure 1b, there is a closed waveguide mode of the red line, which is outside the light line, excluding the well-known bounded SSP mode on the structure. This new leaky mode has also been talked about in a previous work under the name of a cavity mode [38,39] and can open new pathways to design novel plasmonic devices, including leaky wave antenna and fast-wave vacuum electronic devices. In the relevant calculations, the metallic grating parameters are as follows: *a*/*d* = 0.5, *h*/*d* = 2.2, *g*/*d* = 2.8, *d* = 30 μm, λ = 300 μm and *ω*_0_ = 2*π* × 1 THz. Previous studies also indicate that the SSP mode on this roofed metallic grating can realize ultra-high Q resonant tunneling with enhanced emissivity, which may be of great importance to some SSP-based functional devices and systems such as compact and integrated optical sources [40]. 

The metallic grating parameters such as groove depth and width can influence SSP dispersion, which has been largely studied in some previous works but mostly limited to bare metallic gratings without cladding, as shown in Figure 1a [39,40,41,42,43,44,45]. Here, we investigate the influence of gap size on the SSP dispersion change as expressed in Equation (8). SSP dispersion lines of both the bounded mode and leaky mode with different gap sizes of *g* are plotted in Figure 2a. It can be seen that the cutoff frequency of the leaky SSP mode with *k_z_* = 0 gradually shifts to a higher frequency band with a decreased gap size. On the other hand, the propagation constant of the bounded SSP mode becomes larger as the gap size decreases. In addition, the asymptotic frequency of the bounded SSP mode is almost unchanged. This distinct SSP dispersion mode variation with different structural parameters can find different applications that are dependent on the bounded or leaky wave characteristic of the SSP mode. Furthermore, the electric field distributions of the bounded SSP mode on the roofed metallic grating with different gap sizes near asymptotic frequency are also modeled using the finite integration method and are presented in Figure 2b–d with *g* = 15, 30 and 84 μm, respectively. A small gap size between the grating and metal plate can pose a great influence on the propagation of the SSP mode and thus needs to be studied specifically for some SSP-based functional devices and systems. Next, we will investigate the efficient SSP excitation on the roofed metallic grating with a moderately small gap size using an injected electron beam and also give various SSP output power spectrums with different structural or electron beam parameters. 

## 3. THz SSP Excitation on the Roofed Metallic Grating by Electron Beam

Based on above studies on SSP dispersion on the roofed metallic grating, a high-power THz radiation source based on the effective excitation of the bounded SSP mode by an injected electron beam is modeled and studied via particle-in-cell simulations based on the finite-difference-time-domain (FDTD) algorithm [35,36,37,50,51,52,53,54,55,56]. The software studio is also used for some traditional vacuum electronic devices such as backward-wave oscillators (BWO) [51], travelling-wave tubes (TWT) [52], klystrons [53,54], and so on. The working mechanism is similar to some previous works on open space metallic gratings, which is based on the matched dispersion relation between the SSP mode and electron beam [36]. Its schematic diagrams or the proposed functional device of THz SSP excitation by an electron beam on the structure are plotted in Figure 3a. The electron beam is emitted from the left side of the roofed metallic grating with a distance from the grating surface that is marked by H. The total interaction length of the roofed metallic grating structure is L. As shown in Figure 2, the gap size *g* = 30 μm; thus, its dispersion property of the SSP mode is determined. The corresponding dispersion lines of the SSP mode and injected electron beam are plotted in Figure 3b. It can be seen that the electrons are synchronous with the forward wave of the SSP mode. In the simulations, a 2D model is used and the boundary effect along *y* is ignored. Metal is assumed to be a perfect electric conductor and the propagation losses of the SSP mode are ignored [34,35,36,37]. The boundary condition around the interaction system is an absorbing boundary. The dispersion of the injected electron beam is given by *v_b_* = *β* * *c*, *β* = (1 − (1 + γ)^−2^)^0.5^, γ = *U*/*U_r_*(*U_r_* = 5.11 × 10^5^ V is electron energy), where *U* is the beam voltage and *c* is the light velocity. In order to realize a good dispersion match between the SSP mode and electron beam, the velocity of the electron beam *v_b_* can be tuned according to its beam voltage and so the operation frequency is also set. Here, the beam voltage is first set as 18.40 kV and the operation frequency is near 1 THz. Then, its output SSP power spectrums with different beam voltages will be studied and presented. Previous studies show that the output SSP is very sensitive to the distance of the electron beam above the metallic grating of *H* on the open space metallic grating [36]. Here, we also examine the influence of this beam height on the generated SSP power at the end of the structure. Figure 4 gives the simulated SSP output power distributions along a beam height range from 3.4 to 4.0 μm for different interaction lengths of 2.7, 4.0 and 7.6 mm. The obtained SSP power is calculated by integrating the flux along the *x* direction from the bottom to the top of the roofed metallic grating. A magnetic field with 1 T along the *z* direction is used in the system to ensure good electron beam transportation. It can be noted that the effective SSP output happens only with a very limited beam height of *H* and decreases quickly outside this height range. There is almost no output power when the distance is smaller than 3.5 μm or larger than 4 μm. The reason for this is that the interaction between the SSP field and electron beam is not optimal as the field of this specific SSP mode is weak when this distance is very small. Also, it decays rapidly away from the metallic grating surface when the distance is very large. The output SSP power increases gradually along with the interaction waveguide length to an extent. The peak SSP power is reached around 40 W at the end of the structure with a 7.6 mm length for the operation frequency of 1 THz with 18.40 kV and a 0.4 A electron beam. Therefore, the beam height is set as this optimized value and the interaction length is chosen as 7.6 mm for its good output performance. The beam width of emissive shape is also optimized with a half duty cycle and beam width of 3.75 μm. The injected electron beam current is 0.4 A. The interaction length here is largely extended compared to previous studies on the open space metallic grating; thus, it is possible to realize long-distance excitation, provided that the other parameters are properly set [36].

In order to further illustrate the SSP excitation mechanism induced by the electron beam, which is dependent on the SSP dispersion and electron beam energy, the influences of beam energy and the gap size of *g* on the generated SSP power are specifically studied and analyzed in simulations. Figure 5a presents the SSP output power spectrums with different electron beam voltages ranging from 18.10 to 18.70 kV with a 0.1 kV step. The SSP power spectrum, which is plotted by the dependency between the output power and operation frequency, is directly calculated from SSP without an input driving signal. The SSP power spectrum width is about 1 GHz within a 3 dB output level. The obvious red shift of the generated SSP power spectrum is observed with the increased beam voltage. This is caused by the increased electron beam velocity with an increased beam voltage; thus, the optimized operation frequency also decreases according to the above-mentioned beam voltage formulism. The beam current is 0.4 A and the other parameters are kept the same. In the middle of the red line with 18.40 kV, the optimized operation frequency is 1 THz. In addition, the peak value of the generated SSP power slightly decreases along with the increased beam voltage. Thus, the electron beam voltage can provide a simple tuning method to tune the generated SSP power spectrum freely. The SSP power is generated from the decreased electron beam energy across the interaction length. Figure 5b plots the electron beam energy variation along with the interaction length for three different beam voltages of 18.1 (blue line), 18.4 (red line) and 18.7 kV (green line), respectively. It can be clearly seen that the electron beam energy decreases gradually along with the interaction length; thus, the generated SSP power increases at its corresponding optimized operation frequency increases, i.e., 0.9995 THz, 1.0 THz and 1.0005 THz, respectively. In addition, the maximum nest decreased electron beam energy for these three different beam voltages at the end of the structure is about *U* * *I* = 0.3 × 0.4 × 1000 = 120 W; *I* is the injected electron beam current. Due to some diffraction losses and ohmic attenuations, the decreased electron beam energy cannot be fully converted to SSP power. So, the conversion efficiency from the injected electron beam energy to the generated SSP power is about 40/(18,400 × 0.4) = 0.54%.

## 4. Influence of Gap Size on the SSP Excitation

The above studies indicate that the interaction between the SSP mode and injected electron beam is similar to that on the open space grating [36]. To demonstrate its difference on the roofed metallic grating, the gap size of *g* in Figure 3 is specifically considered for its transmission property and the generated SSP power spectrum with the other conditions kept constant. Figure 6 gives the simulated transmission spectrums with different gap sizes of the roofed metallic grating ranging from 10 (blue line) to 50 μm (green line). The inset shows the model in the simulations. The black line and red line are the results for *g* = 20 and 30 μm, respectively. The band-stop property is observed for this periodical roofed metallic grating. In addition, the band-stop window within −2 dB becomes larger as the gap size decreases gradually, as indicated by the black arrow. This distinct transmission property agrees well with the dispersion characteristic variation in Figure 2a. The band-stop window in the transmission is caused by the spectrum gap between the asymptotical frequency of the bounded SSP mode and cutoff frequency of the leaky mode. Obviously, the leaky SSP mode dispersion change mainly contributes to this band-stop window shift as the gap size poses a weak influence on the bounded SSP mode, as shown in Figure 2a. 

To demonstrate the effect of gap size on the generated SSP power spectrums with the roofed metallic grating, three different gap sizes of *g* = 25, 30 and 35 μm were employed to illustrate its interaction mechanism. The other conditions were set the same as each other. Figure 7 provides the simulated results of the generated SSP power spectrums with different gap sizes of *g* = 25, 30 and 35 μm taken from Figure 3, respectively. The electron beam parameters are 18.40 kV and 0.4 A. The metallic grating length is 7.6 mm. It can be concluded that the generated SSP power spectrum shifts gradually to a lower frequency band as the gap size decreases with the same injected electron beam. This variation agrees well with previous dispersion characteristic and transmission property analysis with different gap sizes of the structure. In addition, the peak value of the generated SSP power spectrum decreases along with the decreased gap size. This is because transmission becomes worse as shown in Figure 6 as the gap size decreases; thus, the peak power of SSP also slightly decreases. Changing the gap size between the metal plate and grating can tune the output SSP power spectra and thus should be considered carefully. 

## 5. Conclusions

In this paper, a compact and high-power THz radiation source based on the excitation of the SSP mode on a roofed metallic grating by an electron beam is numerically demonstrated and investigated. The effect of the fundamental dispersion theory on the structure is presented analytically and also compared with that on the conventional bare periodical structure. The excitation of the SSP mode on the structure is also investigated and analyzed by particle-in-cell simulations with various interaction conditions such as different beam energies, height and waveguide lengths, etc. The effects of gap size between the metallic grating and metal plate on the SSP transmission and its generated power spectrum are also examined and studied. The presented studies can provide a new path to develop compact and high-power THz radiation sources induced by an electron beam on the roofed metallic plasmonic waveguide. 

## Figures and Tables

**Figure 1 micromachines-15-00293-f001:**
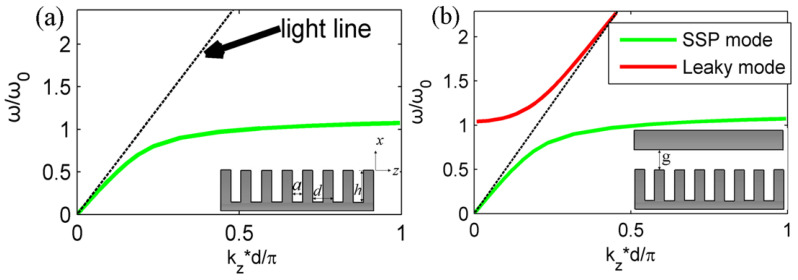
(**a**) SSP dispersion line on the conventional metallic grating with open space boundary. The inset is a detailed parameter of the rectangular metallic grating. The black line is the light line. (**b**) SSP dispersion lines on the considered roofed metallic grating for this study. The green line is the bounded SSP mode and red line is the leaky wave mode. The inset is a structure illustration with a metal plate in the vicinity of corrugated metallic surfaces with a gap size of *g* in between.

**Figure 2 micromachines-15-00293-f002:**
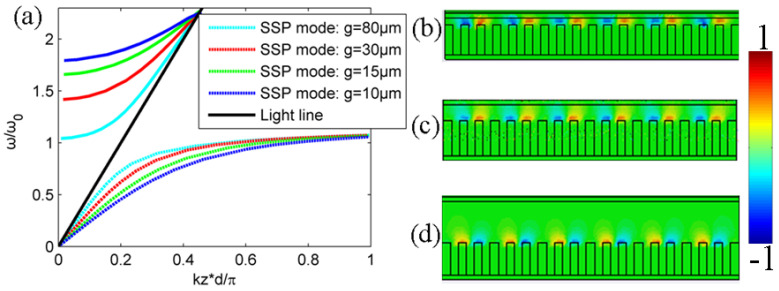
(**a**) SSP dispersion lines with different gap sizes of *g* in the inset of Figure 1b. (**b**–**d**) The normalized electric field distributions of the bounded SSP mode on the roofed metallic grating with different gap sizes of *g* = 15, 30 and 80 μm, respectively.

**Figure 3 micromachines-15-00293-f003:**
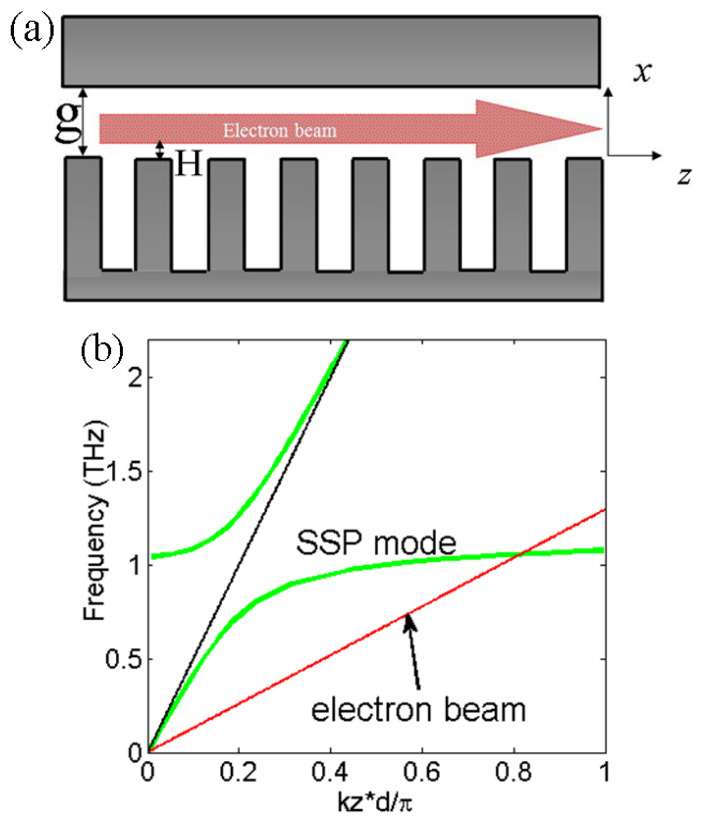
(**a**) Schematic diagrams of the proposed THz SSP excitation on the roofed metallic grating by an injected electron beam on the *x*-*z* plane. The gap size of the metal plate above the grating is marked by *g*. The height of the electron beam is marked by *H*. (**b**) The dispersion lines of the SSP mode and electron beam in the forward wave regime.

**Figure 4 micromachines-15-00293-f004:**
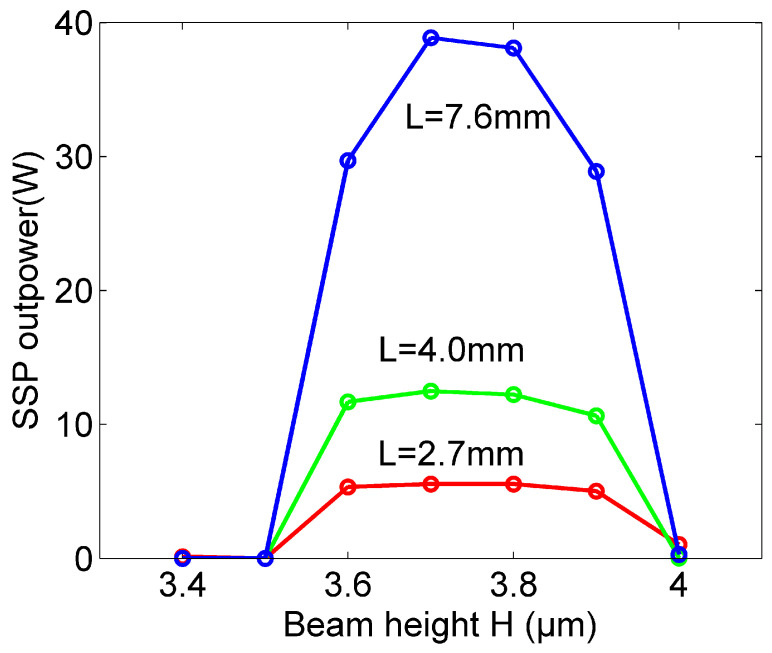
Output SSP power distributions along the electron beam height of *H* with different interaction lengths of 2.7, 4.0 and 7.6 mm from Figure 3, respectively. Beam voltage and current are 18.40 kV and 0.4 A, respectively.

**Figure 5 micromachines-15-00293-f005:**
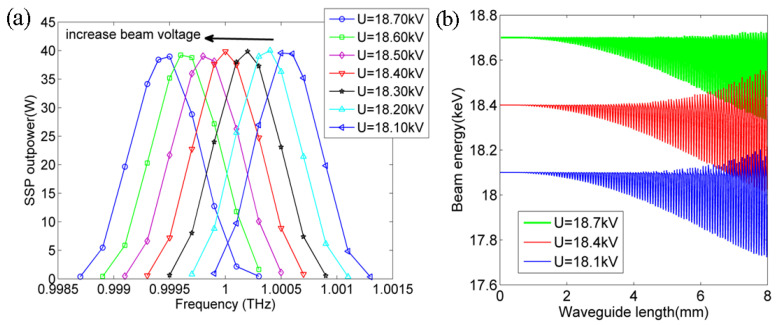
(**a**) Output SSP power spectrums with different electron beam voltages ranging from 18.10 kV to 18.70 kV with a beam current of 0.4 A. The interaction length is 7.6 mm. (**b**) The electron beam energy variations along with the interaction length for different beam voltages of 18.1, 18.4 and 18.7 kV at its corresponding optimized operation frequency, respectively.

**Figure 6 micromachines-15-00293-f006:**
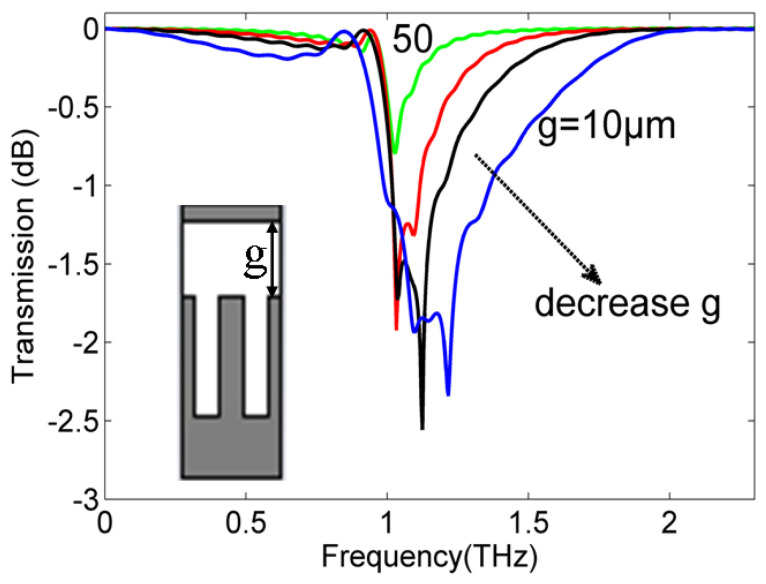
The transmission magnitude distributions of the roofed metallic grating with different gap sizes between the metallic grating and metal plate ranging from 10 to 50 μm. The inset is the calculated model in the simulations.

**Figure 7 micromachines-15-00293-f007:**
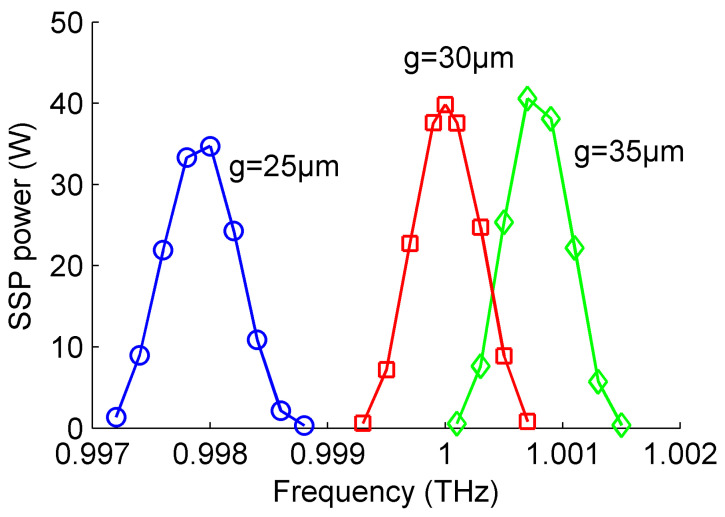
The generated SSP power spectrums with different gap sizes of *g* = 25, 30 and 35 μm taken from Figure 3, respectively. The electron beam parameters are 18.40 kV and 0.4 A. The metallic grating length is 7.6 mm.

## Data Availability

The data presented in this study are available on request from the corresponding author.

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
