# Peer review of "Excitation of Terahertz Spoof Surface Plasmons on a Roofed Metallic Grating by an Electron Beam"

_micromachines, 2024, doi:10.3390/mi15030293_

Round 1

Reviewer 1 Report

Comments and Suggestions for Authors

This paper presents a thorough examination and investigation of the fundamental SSP modes on roofed metallic gratings and their effective excitation by an injected electron beam in the THz regime. The introduction of the roofed metallic grating allows for the generation of a closed waveguide mode outside the light line, while the bounded SSP mode line shifts to lower frequencies. The effective excitation of the bounded SSP mode on the roofed metallic grating is studied using a particle-in-cell simulation studio, and the resulting SSP power spectra with various gap sizes are obtained and analyzed. The simulation results indicate a slight red shift in the generated SSP spectra with decreasing gap size. This research on the excitation of SSP mode by an electron beam has implications for the development of high-power compact THz radiation sources, leveraging the strong near-field confinement of SSPs on metallic gratings.

I think this work is interesting and should be accepted asap.

Comments on the Quality of English Language

English need be more simple

Author Response

Thanks for you comments.

Reviewer 2 Report

Comments and Suggestions for Authors

As the reported paper is purely simulation, authors should reflect this in the title by adding "Theoretical Study" or "Simulation" words.

In Figure 2 (a) insert, g=80um is written. In the Figure caption, it mentioned 84um. Which one is the correct?

I have no other comments other than the above 2.

Author Response

Thanks for your report.

We have changed 84um to 80 in the figure caption.  The title is too long if we add "simulated study". So, in the Abstract, "numerically" is added according to your suggestions.